

# Comprehensive assessment of the coupling coordination degree between urbanization and ecological environment in the Siberian and Far East Federal Districts, Russia from 2005 to 2017

Ji Zheng[1,2], Yingjie Hu[3], Tamir Boldanov[1,2], Tcogto Bazarzhapov[1,2], Dan Meng[1,2], Yu Li[1,2] and Suocheng Dong[1,2]

[1] Institute of Geographic Sciences and Natural Resources Research, Chinese Academy of Sciences, Beijing, China
[2] University of Chinese Academy of Sciences, Beijing, China
[3] College of City Construction, Jiangxi Normal University, Nanchang, China

Corresponding authors
Yu Li, liy@igsnrr.ac.cn
Suocheng Dong,
dongsc@igsnrr.ac.cn

## ABSTRACT

The urbanization growth in the 20th and 21st centuries has led to a series of unprecedented problems in the ecological environment. Based on constructing an integrated urbanization-ecological environment index system, this article conducts a comprehensive evaluation of the coupling coordination degree between urbanization and the ecological environment and uncovers its spatiotemporal variation characteristics in the Siberian and Far East Federal Districts, Russia from 2005 to 2017. The coupling coordination of urbanization and the ecological environment in the Siberian and Far East Federal Districts improve from slightly unbalanced development stage to barely balanced development stage from 2005 to 2017. In 2017, more than half regions achieved the barely balanced development of urbanization and the ecological environment. However, the most desirable development stage, the superior balanced development stage, is never achieved in the Siberian and Far East Federal Districts during the study period. The spatial pattern of the coupling coordination degree of urbanization and the ecological environment in the Siberian and Far East Federal District gradually changes from "dumbbell" to "high-north low-south". The south part of the Siberian and Far East Federal Districts should be paid more attention in the future urban development process. This research will provide support in the future coordination of urban development in the Siberian and Far East Federal Districts.

## INTRODUCTION

Urbanization is a process that comprehensively shifts from rural to urban, transforming their population, land, economies, and social properties (*Bekhet & Othman, 2017*). Since 2008, the proportion of the global population living in urban areas is over 50%

(*United Nations, 2019*). During the 20th and 21st centuries, urbanization in developing countries has occurred at unprecedented rates. While producing enormous benefits in employment, infrastructure, service, welfare and innovation (*Ochoa et al., 2018*), urbanization has also caused a series of ecological and environmental problems (*Zhang, 2016*), including PM2.5-dominated air pollution (*Fang et al., 2015*; *Du et al., 2019*), water pollution (*Ma, Chou & Wang, 2016*), climate change (*Gurney et al., 2015*) and health challenges (*Gong et al., 2012*). Therefore, to achieve sustainable urban development, it is necessary to explore the relationship between urbanization and the ecological environment.

Many studies have provided important insights on the relationship between urbanization and the ecological environment. There are various methods being applied to investigate the relationship between urbanization and the ecological environment: Stochastic Impacts by Regression on Population, Affluence and Technology (STIRPAT) model (*Lin et al., 2017*), Kaya Identity (*Zhang et al., 2017b*), Logarithmic Mean Divisia Index (LMDI) method (*Ding & Li, 2017*), Ordinary Least Square (OLS) method (*Yu et al., 2018*), Geographically Weighted Regression (GWR) model (*Liang, Wang & Li, 2019*), panel data regression (*Yao et al., 2018*; *Du & Xia, 2018*), Grey Correlation Analysis (*Zhang et al., 2017a*), Environment Kuznets Curve (EKC) approach (*Xu, Dong & Yang, 2018*), Granger Causality Test (*Bekhet & Othman, 2017*; *Ahmed, Wang & Ali, 2019*) and Coupling Coordination Degree Method (CCDM) (*Song et al., 2018*; *Geng et al., 2020*). However, the STIRPAT model, Kaya Identity, LMDI method, OLS method, GWR model, and panel data regression can only reveal the unidirectional influences of urbanization on the ecological environment. Grey Correlation Analysis, EKC approach, Granger Causality Test and CCDM can enrich the understanding of the bidirectional relationship between urbanization and the ecological environment. The Grey Correlation Analysis can reveal the synchronization degree of the variation of urbanization and the ecological environment and does not require a large-size sample. The EKC approach can express the relationship between urbanization and the ecological environment clearly and it is easy to use. The Granger Causality Test can reveal the influence of urbanization on the ecological environment, and it can also unravel the response of the ecological environment to urbanization. However, the Grey correlation analysis, EKC approach and Granger Causality Test cannot reveal the spatiotemporal variation characteristics of the relationship between urbanization and the ecological environment. From the perspective of coordinated development, the CCDM can assess the coordination level between urbanization and the ecological environment and reveal the spatiotemporal variation characteristics of the coordination development of urbanization and the ecological environment. *Fang, Liu & Li (2016)* proposed that the coupling relationship between urbanization and the eco-environment will become an important international research frontier over the next 10 years. Recently, the CCDM method was widely applied to investigate the relationship between urbanization and the ecological environment (*Guo et al., 2015*; *Liu et al., 2018*; *Xu & Hou, 2019*). For example, *He et al. (2017)* examined the relationship between urbanization and the eco-environment in Shanghai from 1980 to 2013. *Yao et al. (2019)* calculated the degree of coupling coordination between urbanization and ecological environments at the provincial level in 2005, 2010 and 2015 in

China and observed obvious regional disparity. However, most existing research was conducted in China. Only *Dong et al. (2019)* conducted a quantitative assessment of the coupling coordination degree of urbanization and the eco-environment in Mongolia at the provincial level, and *Zhao et al. (2017)* investigated the relation of 2019 countries and regions all over the world in 2014.

Russia is an important country in the China–Mongolia–Russia Economic Corridor. To promote the national and international influences of the Siberian and Far East region, the Russian federal government proposed the Strategy of Socioeconomic Development of the Far East and the Baikal Region until 2025. In 2018, Russia and China jointly signed the China-Russia Cooperation and Development Plan in the Far East (2018–2024) to prompt the development of the Far East. The Siberian and Far East Federal Districts are expected to develop rapidly and face more challenges in their ecological environment caused by more severe anthropogenic activities over the next decade. An assessment of the coupling coordination degree of urbanization and the ecological environment in the Siberian Federal District and Far East Federal District is urgently needed. However, due to data limitations, the related research is still somewhat insufficient in this region. *Chu et al. (2018)* measured urbanization in the Siberian and Far East Federal Districts in Russia from the perspectives of population, economy and society, and revealed the spatial heterogeneity of urbanization development. *Zhu et al. (2018)* investigated the influences of urbanization and income on $CO_2$ emissions in the BRICS (Brazil, Russia, India, China and South Africa) and observed a significantly negative influence of urbanization on $CO_2$ emissions. *Fan et al. (2018)* explored the relationship between urbanization and sustainability in Asian Russia from 1990 to 2014 and found a generally positive trend of urbanization during the study period.

To rectify the above shortcomings, this study establishes a comprehensive index system for the urbanization-ecological environment system, quantitatively evaluates the coupling coordination degree of urbanization and the ecological environment, and reveals the spatiotemporal variations of the coupling coordination degree in the Siberian and Far East Federal Districts in Russia during 2005–2017. It will provide scientific support and insights into coordinating urbanization development and ecological environment protection in the Siberian and Far East Federal Districts in Russia.

## MATERIALS AND METHODS

### Study area and data collection

This study conducts a comprehensive evaluation of the coupling coordination degree of urbanization and ecological environment in the Siberian Federal District and the Far East Federal District (Fig. 1). The Siberian Federal District includes three republics (the Republic of Altay, Republic of Tuva, and Republic of Khakassia), five oblasts (Omsk Oblast, Novosibirsk Oblast, Tomsk Oblast, Kemerovsk Oblast and Irkutsk Oblast), and two krais (Altai Kray and Krasnoyarsk Kray). The Far East Federal District includes two republics (the Republic of Buryatia and Republic of Sakha), three oblasts (Amursk Oblast, Magadansk Oblast and Sakhalin Oblast), two autonomous oblasts (Jewish Autonomous Oblast and Chukotska Autonomous Oblast), and four krais (Khabarovskiy Kray, Primorskiy Kray, Kamchatskiy Kray and Zabaykalskiy Kray). All urbanization and ecological environment

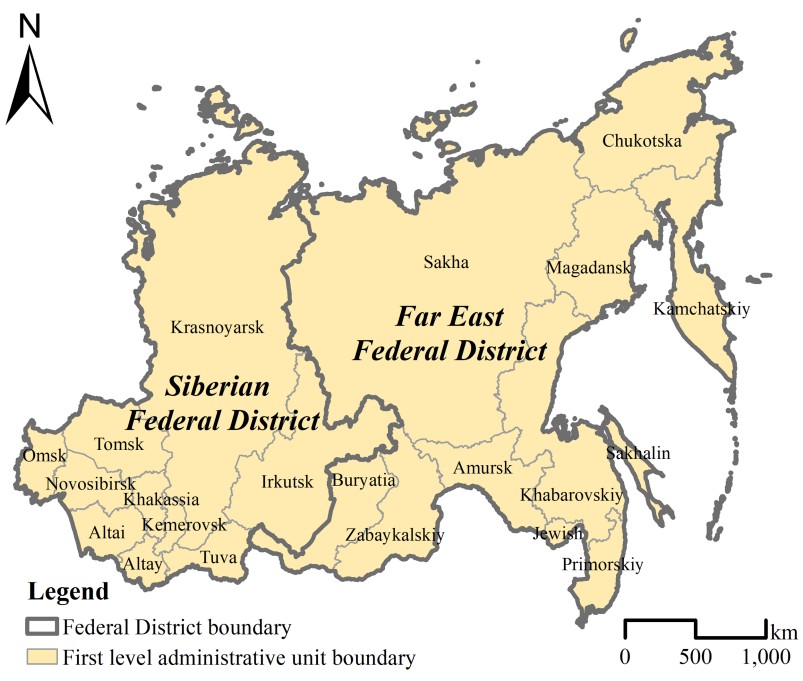

**Figure 1** Administrative division of the Siberian and Far East Federal Districts.

data from 2005 to 2017 were collected from the Regions of Russia Socio-Economic Indicators P32 Statistical Yearbook (*Russian Federal State Statistics Service, 2006*).

## Comprehensive urbanization-ecological environment index system

Urbanization is an integrated development that not only reflects an increase in urban population but also an expansion of urban land, economy and lifestyle to a rural area (*Weng, 2007*; *Bekhet & Othman, 2017*). Considering the comprehensiveness of the process of urbanization and the data limitation of spatial urbanization in the Siberian and Far East Federal Districts, three first grade indicators, including demographic urbanization, economic urbanization and social urbanization, are selected into the urbanization subsystem. Referred to the previous research, which build the urbanization index system and investigate the relation between urbanization and ecological environment (*Zhao, Wang & Zhou, 2016*; *He et al., 2017*; *Wang, Gao & Li, 2020*), the most cited and as comprehensive as possible basic indicators are collected. Therefore, the urbanization index subsystem consists of three primary indicators and 12 basic indicators (Table 1). According to the environmental pressure-state-response framework, widely used in the ecological environment (*Hughey et al., 2004*; *Liu et al., 2018*), we establish an ecological environment index subsystem containing three first grade indicators (ecological environment pressure, ecological environment state, and ecological environment response). Regarding the basic indicators of the ecological environment pressure and response, the solid, liquid and air pollutants are covered as much as possible. Since the crops and forests are both essential ecological and resource elements, forest area per capita and sown area of all crops per capita are selected as the basic indicators of the ecological environment state.

**Table 1 Urbanization-ecological environment index system.**

| Index system | First grade indicator | Weight (%) | Basic indicator | Weight (%) |
|---|---|---|---|---|
| Urbanization subsystem | Demographic urbanization | 28.34 | Percentage of urban population (%) | 2.53 |
| | | | Percentage of economic activity population (%) | 3.62 |
| | | | Population density (people/km$^2$) | 22.19 |
| | Economic urbanization | 32.07 | Gross regional product per capita (rubles) | 14.54 |
| | | | Average income per capita (rubles/month) | 8.88 |
| | | | Per capita monetary expenses and savings (rubles) | 7.15 |
| | | | Unemployment rate (%) | 1.50 |
| | Social urbanization | 39.59 | Number of high education institutions per 10,000 population | 5.99 |
| | | | Number of doctors per 10,000 population | 5.62 |
| | | | Number of sports facilities per 10,000 population | 14.53 |
| | | | Number of public busses per 100,000 population | 7.36 |
| | | | The volume of communication services per capita | 6.09 |
| Ecological environment subsystem | Ecological environment state | 55.64 | Forest area per capita (ha) | 29.22 |
| | | | Sown area of all crops per capita (ha) | 26.42 |
| | Ecological environment pressure | 6.33 | Air pollutants emitted from stationary sources per capita (kg) | 2.50 |
| | | | Polluted water discharged into surface water per capita (m$^3$) | 2.76 |
| | | | Solid waste per capita (kg) | 1.08 |
| | Ecological environment response | 38.03 | Capture of air pollutants from stationary sources per capita (kg) | 19.28 |
| | | | Volume of circulating and consistency used water per capita (m$^3$) | 18.75 |

We employed the modified entropy method to determine the weight of each indicator in the comprehensive urbanization-ecological environment index system. The detailed steps of the entropy model are as follows:

Normalization pre-processing. All indicators were normalized by Eqs. (1) and (2) to remove the effects of dimension, magnitude and orientation:

Positive indicator (favorable condition):

$$r_{ij} = \frac{x_{ij} - \min(x_j)}{\max(x_j) - \min(x_j)} \tag{1}$$

Negative indicator (unfavorable condition):

$$r_{ij} = \frac{\max(x_j) - x_{ij}}{\max(x_j) - \min(x_j)} \tag{2}$$

where $i$ denotes the year, $j$ denotes the indicator, $r_{ij}$ denotes the normalized value, $X_{ij}$ denotes the original value, $\max(X_j)$ denotes the maximum value of the indicator $j$ during the study period, and $\min(X_{ij})$ denotes the minimum value of the indicator $j$ during the study period.

Information entropy of the indicator $j$ ($e_j$):

$$e_j = -\frac{1}{\ln n}\sum_{i=1}^{n} X_{ij} \times \ln X_{ij}(0 \leq e_j \leq 1) \tag{3}$$

$$X_{ij} = \frac{r_{ij}}{\sum_{i=1}^{n} r_{ij}} \tag{4}$$

where $X_{ij}$ denotes the proportion of the indicator $j$ in year $i$, $n$ is the number of years (2005–2017, $n = 13$).

Entropy redundancy ($g_j$):

$$g_j = 1 - e_j \tag{5}$$

Weight of the indicator $j$ ($w_j$):

$$w_j = \frac{g_j}{\sum_{j=1}^{m} g_j} \tag{6}$$

where $m$ denotes the number of indicators ($m = 11$ for the urbanization index subsystem, $m = 6$ for the ecological environment subsystem).

Evaluation of the indicator ($Y_{ij}$):

$$Y_{ij} = w_j \times r_{ij} \tag{7}$$

Comprehensive level in year $i$ ($Y_i$):

$$Y_i = \sum_{j=1}^{m} Y_{ij} \tag{8}$$

The weight of each indicator in the urbanization-ecological environment index system is shown in Table 1.

## Coupling coordination degree model

Coupling is the interactive influence between two or more systems. Urbanization may cause a series of ecological and environmental problems (*Chen et al., 2006*), and conversely, a deteriorated and fragile ecological environment will limit urbanization development (*Li et al., 2012*). The modified coupling coordination degree model is as follows (*He et al., 2017*):

$$C = \{f(U) \times g(E)/([f(U) + g(E)]/2)^2\}^{1/2} \tag{9}$$

where $C$ denotes the coupling degree of urbanization and ecological environment, $f(U)$ denotes the urbanization subsystem, and $g(E)$ denotes the ecological environment subsystem.

The level of influence of urbanization and ecological environment:

$$T = \alpha f(U) + \beta g(E) \tag{10}$$

where $T$ denotes the level of influence of urbanization and ecological environment, $\alpha$ denotes the contribution of urbanization to the comprehensive system, and $\beta$ denotes the

**Table 2 Typology of the development stages of the coupling coordination degree of urbanization and ecological environment.**

| Primary stages | | Basic stages | |
| --- | --- | --- | --- |
| Superior balanced development | $0.8 < D \leq 1$ | $g(E) - f(U) > 0.1$ | Superiorly balanced development with lagging urbanization |
| | | $f(U) - g(E) > 0.1$ | Superiorly balanced development with a lagging ecological environment |
| | | $0 \leq |f(U) - g(E)| \leq 0.1$ | Superiorly balanced development of urbanization and ecological environment |
| Barely balanced development | $0.5 < D \leq 0.8$ | $g(E) - f(U) > 0.1$ | Barely balanced development with lagging urbanization |
| | | $f(U) - g(E) > 0.1$ | Barely balanced development with a lagging ecological environment |
| | | $0 \leq |f(U) - g(E)| \leq 0.1$ | Barely balanced development of urbanization and ecological environment |
| Slightly unbalanced development | $0.3 < D \leq 0.5$ | $g(E) - f(U) > 0.1$ | Slightly unbalanced development with hindered urbanization |
| | | $f(U) - g(E) > 0.1$ | Slightly unbalanced development with a hindered ecological environment |
| | | $0 \leq |f(U) - g(E)| \leq 0.1$ | Slightly unbalanced development of urbanization and ecological environment |
| Seriously unbalanced development | $0 < D \leq 0.3$ | $g(E) - f(U) > 0.1$ | Seriously unbalanced development with hindered urbanization |
| | | $f(U) - g(E) > 0.1$ | Seriously unbalanced development with a hindered ecological environment |
| | | $0 \leq |f(U) - g(E)| \leq 0.1$ | Seriously unbalanced development of urbanization and ecological environment |

ecological environment's contribution to the comprehensive system. The value of α and β are also determined by entropy method (α = 0.467, β = 0.533).

The coupling coordination degree of urbanization and ecological environment:

$$D = \sqrt{C \times T} \tag{11}$$

Table 2 shows the typology of the developmental stages of the coupling coordination degree of urbanization and the ecological environment. It can be divided into four primary stages (balanced development, transitional development, slightly unbalanced development, and seriously unbalanced development) and 12 basic stages.

## RESULTS

Each indicator's weight in the urbanization index subsystem and the ecological environment index subsystem is assessed by the entropy method separately (Table 1). The weights of the basic indicators of the urbanization index subsystem are: population density (22.19%) > gross regional product per capita (14.54%) > number of sports facilities per 10,000 population (14.53%) > average income per capita (8.88%) > per capita monetary expenses and savings (7.15%) > the volume of communication services per capita (6.09%) > number of high education institutions per 10,000 population (5.99%) > number of doctors per 10,000 population (5.62%) > percentage of economic activity population (3.62%) > unemployment rate (1.50%). Three basic indicators, population density, gross regional product per capita and number of sports facilities per 10,000 population, account for more than half of the total impact in relation to the

comprehensive urbanization level. Our research suggests that population aggregation, economic growth and facilities improvement are all the important driving factors of the urbanization development of the Siberian and Far East Federal Districts during the past decade. This result is somewhat different from existing research. *He et al. (2017)* and *Liu et al. (2018)* found that percentage of urban population, per capita GDP and economic structure have great influence on the comprehensive urbanization in China in the recent decades. *Dong et al. (2019)* found that population density and economic structure are key driving elements of the comprehensive urbanization development in Mongolia. Since the high percentage of urban population of most regions in the Siberian and Far East Federal Districts (only the urban population rate of the Republic of Altay is below 50%, at 29%), population density, not percentage of urban population, has great influence on the comprehensive urbanization level. There are two main reasons for the high percentage of urban population in the Siberian and Far East Federal Districts: (1) More and better work opportunities attract people to aggregate in urban areas; and (2) higher level of the infrastructure and services in the urban area. In particular, heating costs in this severely cold region are much higher in rural areas than urban areas. Our results support that for the sparsely populated and severely cold Siberian and Far East Federal Districts, policy makers should pay more attention to the population density and the construction of urban infrastructures and service facilities when formulating the urban development policies.

A total of 4 basic indicators, including forest area per capita (29.22%), sown area of all crops per capita (26.42%), capture of air pollutants from stationary sources per capita (19.28%) and volume of circulating and consistency used water per capita (18.75), account for 93.67% of the total influences on the ecological environment. The ecological environment state and ecological environment response are the essential factors of the ecological environment subsystem. The results indicate that the intrinsic endowment heterogeneity of the ecological environment is high in the vast territory, and that existing ecological environment protection measures in the Siberian and Far East Federal Districts are effective.

## Spatial pattern of the comprehensive urbanization level

The comprehensive urbanization scores of the Siberian Federal District and the Far East Federal District are obtained by averaging the comprehensive urbanization scores of each region. The comprehensive urbanization scores of the two federal districts from 2005 to 2017 are shown in Fig. 2. The comprehensive urbanization scores of the Siberian Federal District and the Far East Federal District both show an increasing trend during the period of 2005–2017. The comprehensive urbanization scores of the Far East Federal District increases faster than the Siberian Federal District during the study period. The comprehensive urbanization score of the Siberian Federal District increases from 0.1787 in 2005 to 0.2902 in 2017, with average annual growth rate of 4.12%. The comprehensive urbanization score of the Far East Federal District increases from 0.1650 in 2005 to 0.3541 in 2017, with average annual growth rate of 6.57%. It is noteworthy that 2009 is a turning point. From 2005 to 2008, the comprehensive urbanization score of the Siberian Federal District was higher than the Far East Federal District. Since 2009, the comprehensive

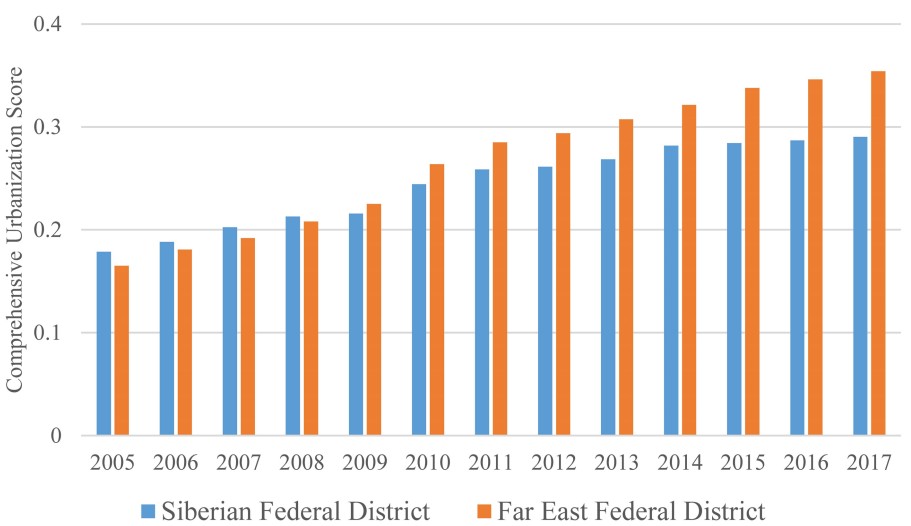

**Figure 2 Comprehensive urbanization scores of the Siberian and Far East Federal Districts.**

urbanization score of the Far East Federal District surpassed the comprehensive urbanization score of the Siberian Federal District and the gap of the comprehensive urbanization level between the Far East Federal District and the Siberian Federal District become larger continuously from 2009 to 2017. In 2009, the *Outline of Cooperation Plan between Northeast Region in China and Far East and East Siberian Region in Russia* was signed by the governments of China and Russia promote the cooperation between the two countries in the fields of trade, investment, infrastructure construction, energy and high technology. In 2014, the government of the Russia enacted the *Federal Law of Russian Social Economic Advanced Development Region* to promote the development of the Far East Federal Districts. These macroscopic policies proposed in the recent decade are important for the relatively rapid urbanization development in the Far East Federal District.

The "dumbbell" spatial pattern of the comprehensive urbanization levels in the Siberian and Far East Federal Districts during 2005–2017 is shown in Fig. 3. The Siberian and Far East Federal Districts have higher comprehensive urbanization levels in the western and eastern regions, and lower comprehensive urbanization levels in the central region. In 2005, the comprehensive urbanization scores are ranked as follows: Kemerovsk Oblast (0.3446) > Novosibirsk Oblast (0.2489) > Omsk Oblast (0.2334) > Chukotska Autonomous Oblast (0.2160) > Khabarovskiy Kray (0.2142) > Altai Kray (0.2019) > Kamchatskiy Kray (0.1971) > Amursk Oblast (0.1933) >Tomsk Oblast (0.1808) > Sakhalin Oblast (0.1800) > Magadansk Oblast (0.1632) > Republic of Khakassia (0.1537) > Primorskiy Kray (0.1477) > Republic of Sakha (0.1469) > Zabaykalskiy Kray (0.1464) > Krasnoyarsk Kray (0.1395) > Irkutsk Oblast (0.1380) > Jewish Autonomous Oblast (0.1219) > Republic of Buryatia (0.0882) > Republic of Tuva (0.0752) > Republic of Altay (0.0713). The top 10 comprehensive urbanization score regions consist of 5 regions in the Siberian Federal District and 5 regions in the Far East Federal District. In 2017, the comprehensive urbanization scores are ranked as follows: Sakhalin Oblast (0.4863) > Primorskiy Kray

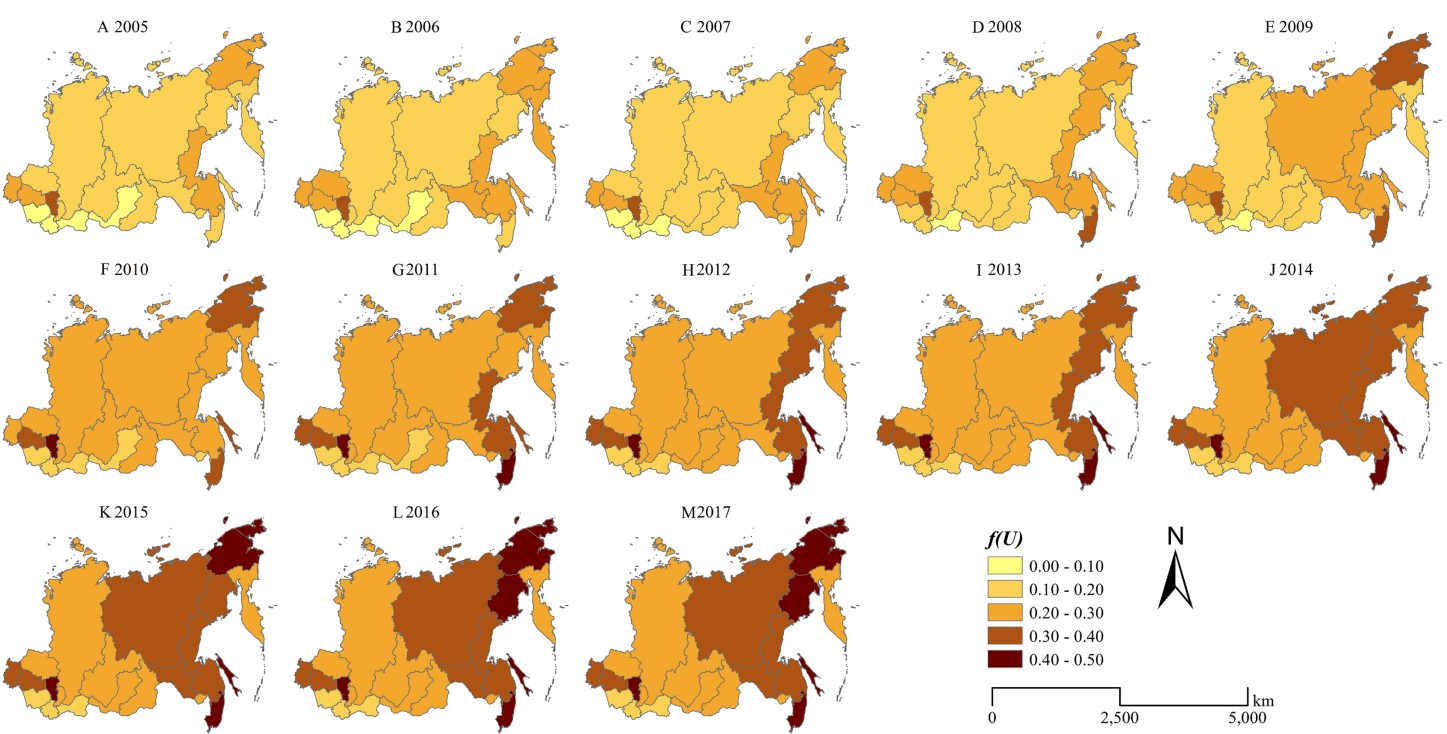

**Figure 3 Spatiotemporal pattern of the comprehensive urbanization level in the Siberian and Far East Federal Districts.** (A) 2005, (B) 2006, (C) 2007, (D) 2008, (E) 2009, (F) 2010, (G) 2011, (H) 2012, (I) 2013, (J) 2014, (K) 2015, (L) 2016, (M) 2017.

(0.4777) > Chukotska Autonomous Oblast (0.4469) > Kemerovsk Oblast (0.4328) > Magadansk Oblast (0.4304) > Novosibirsk Oblast (0.3794) > Omsk Oblast (0.3720) > Khabarovskiy Kray (0.3505) > Republic of Sakha (0.3449) > Amursk Oblast (0.3409) > Altai Kray (0.3009) > Krasnoyarsk Kray (0.2770) > Republic of Buryatia (0.2679) > Republic of Khakassia (0.2668) > Tomsk Oblast (0.2654) > Kamchatskiy Kray (0.2584) > Irkutsk Oblast (0.2555) > Zabaykalskiy Kray (0.2514) > Jewish Autonomous Oblast (0.2395) > Republic of Tuva (0.1907) > Republic of Altay (0.1619). The top 10 comprehensive urbanization score regions consist of 3 regions in the Siberian Federal District and 7 regions in the Far East Federal District in 2017. From 2005 to 2017, the comprehensive urbanization level in every region of the Siberian and Far East Federal Districts all kept a continuous increasing trend. However, the comprehensive urbanization develop more rapidly in the Far East Federal District. During the period of 2005–2017, there are 8 regions of the top 10 fastest urbanization growing regions in the Far East Federal District. It suggests that the development policies for the Far East Federal District proposed by the government of Russian Federation in the past decade make good progress on the social and economic development of the Far East Federal District. Despite the differences of the growing rate the comprehensive urbanization level between the Siberian Federal District and the Far East Federal District, the spatial characteristics of the comprehensive urbanization level in the Siberian and Far East Federal District keep the "dumbbell" pattern.

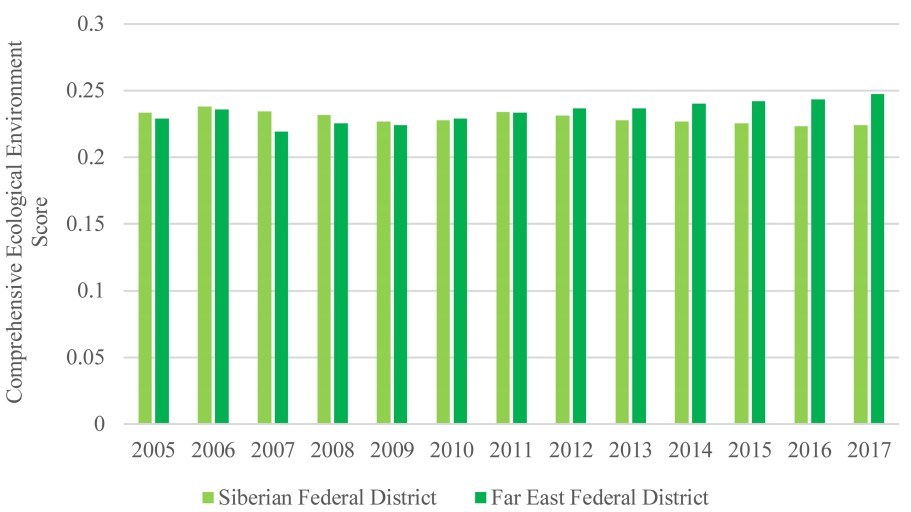

**Figure 4** Comprehensive ecological environment scores of the Siberian and Far East Federal Districts.

## Comprehensive ecological environment level

The comprehensive ecological environment scores of the Siberian Federal District and the Far East Federal District are calculated by averaging the comprehensive ecological environment scores in the three republics, five oblasts, and two krais in the Siberian Federal District, and in the two republics, three oblasts, two autonomous oblasts, and four krais in the Far East Federal District, respectively. The temporal variation characteristics of the comprehensive ecological environment scores of the Siberian Federal District and the Far East Federal District during 2005–2017 are shown in Fig. 4. The comprehensive ecological environment level of the Siberian and Far East Federal Districts are similar and keep stable between 0.2 and 0.25 during the period of 2005–2017. The stable comprehensive ecological environment level in the Siberian and Far East Federal Districts in the recent decade indicates that the environment protection measures taken by the government and public have significantly positive influences.

The comprehensive ecological environment of the Siberian and Far East Federal Districts shows a "high-north low-south" spatial pattern (Fig. 5). The comprehensive ecological environment scores are relatively higher in the north region, where are high latitude and cold. The population of the high latitude and cold region in the north part of the Siberian and Far East Federal Districts are scarce. Richer per capita natural resources and weaker destruction from anthropogenic activities on the ecological environment cause the comprehensive ecological environment scores higher in the north region of the Siberian and Far East Federal Districts. However, the ecological environment of the high latitude and cold region is fragile. It's difficult to recover since the fragile ecological environment is destroyed. Therefore, the north region of the Siberian and Far East Federal Districts should be paid attention during the urbanization process though the north region have relatively higher comprehensive ecological environment scores.

In 2005, the comprehensive ecological environment scores are ranked as follows: Republic of Sakha (0.3999) > Krasnoyarsk Kray (0.3969) > Chukotska Autonomous

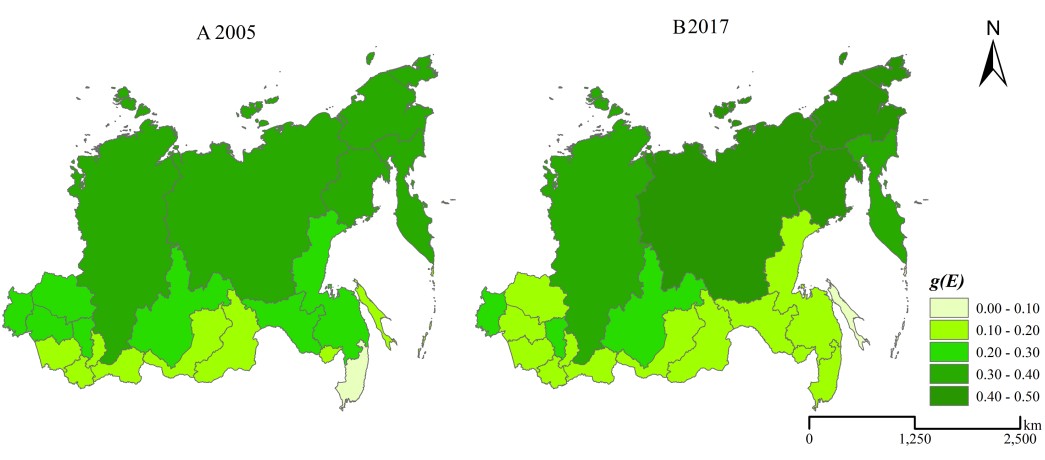

**Figure 5 Spatial pattern of the comprehensive ecological environment in the Siberian and Far East Federal Districts.** (A) 2005, (B) 2017.               

Oblast (0.3799) > Kamchatskiy Kray (0.3371) > Altai Kray (0.3324) > Magadansk Oblast (0.3166) > Omsk Oblast (0.2772) > Tomsk Oblast (0.2567) > Amursk Oblast (0.2549) > Kemerovsk Oblast (0.2491) > Khabarovskiy Kray (0.2142) > Novosibirsk Oblast (0.2064) > Irkutsk Oblast (0.2014) > Zabaykalskiy Kray (0.1639) > Republic of Altay (0.1574) > Republic of Buryatia (0.1422) > Republic of Khakassia (0.1352) > Jewish Autonomous Oblast (0.1264) > Republic of Tuva (0.1242) > Sakhalin Oblast (0.1062) > Primorskiy Kray (0.0794). In 2017, the comprehensive ecological environment scores are ranked as follows: Republic of Sakha (0.4175) > Magadansk Oblast (0.4150) > Chukotska Autonomous Oblast (0.4050) > Altai Kray (0.3530) > Kamchatskiy Kray (0.3483) > Krasnoyarsk Kray (0.3343) > Omsk Oblast (0.2998) > Kemerovsk Oblast (0.2453) > Irkutsk Oblast (0.2134) > Amursk Oblast (0.1945) > Novosibirsk Oblast (0.1942) > Zabaykalskiy Kray (0.1811) > Jewish Autonomous Oblast (0.1764) > Khabarovskiy Kray (0.1733) > Primorskiy Kray (0.1718) > Republic of Khakassia (0.1653) > Tomsk Oblast (0.1611) > Republic of Altay (0.1556) > Republic of Buryatia (0.1394) > Republic of Tuva (0.1184) > Sakhalin Oblast (0.0978). The comprehensive ecological environment scores in most regions of the Siberian and Far East Federal Districts keep stable, with the difference of the comprehensive ecological environment score between 2005 and 2017 lower than 0.03. The comprehensive ecological environment scores of 4 regions, including Khabarovskiy Kray, Amursk Oblast, Krasnoyarsk Kray and Tomsk Oblast, decrease more than 0.03 during the period of 2005–2017. The comprehensive ecological environment scores of 3 regions, including Magadansk Oblast, Primorskiy Kray and Jewish Autonomous Oblast, increase more than 0.05 during 2005 and 2017.

## Development stages and spatiotemporal patterns of the coupling coordination degree between urbanization and the ecological environment

We calculated the coupling coordination degree between urbanization and the ecological environment in the Siberian Federal District and the Far East Federal District using the CCDM model on the basis of the comprehensive urbanization-ecological environment

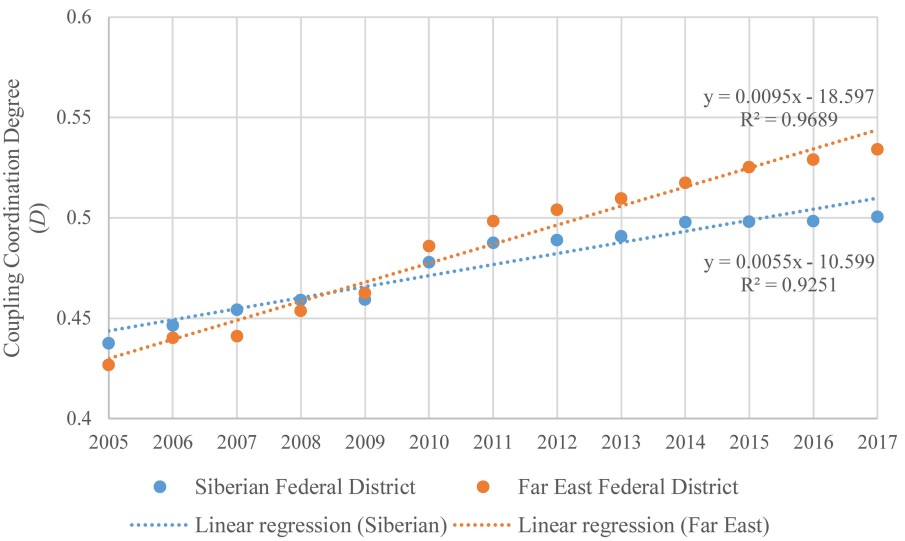

**Figure 6 Temporal variation of the coupling coordination degree in the Siberian and Far East Federal Districts.**

index system. According to the coupling coordination degree, the development is divided into four primary stages: superior balanced development, barely balanced development, slightly unbalanced development, and seriously unbalanced development. Each primary stage is divided into three basic stages by comparing the comprehensive urbanization score to the comprehensive ecological environment score. The coupling coordination degree between urbanization and the ecological environment of the Siberian Federal District and the Far East Federal District are obtained by averaging the coupling coordination degree values of the three republics, five oblasts and two krais in the Siberian Federal District, and the two republics, three oblasts, two autonomous oblasts and four krais in the Far East Federal District, respectively.

The coupling coordination degree of urbanization and the ecological environment in the Siberian and Far East Federal Districts during 2005–2017 is shown in Fig. 6. From 2005 to 2017, the coupling coordination degree of urbanization and the ecological environment in the Siberian Federal District and the Far East Federal District both keep an increasing trend. The coupling coordination degree of urbanization and the ecological environment in the Far East Federal District increases faster than the Siberian Federal District. In the Siberian Federal District, the coupling coordination degree of urbanization and the ecological environment increases from 0.4375 in 2005 to 0.5005 in 2017, with average annual growth rate at 1.13%. The Siberian Federal District achieved the coupling coordination development of urbanization and the ecological environment changing from the slightly unbalanced development stage to the barely balanced development stage in 2017. In the Far East Federal District, the coupling coordination degree of urbanization and the ecological environment increases from 0.4270 in 2005 to 0.5342 in 2017, with average annual growth rate at 1.89%. The Far East Federal District achieved the coupling coordination development of urbanization and the ecological environment changing from the slightly unbalanced development stage to the barely balanced development stage in

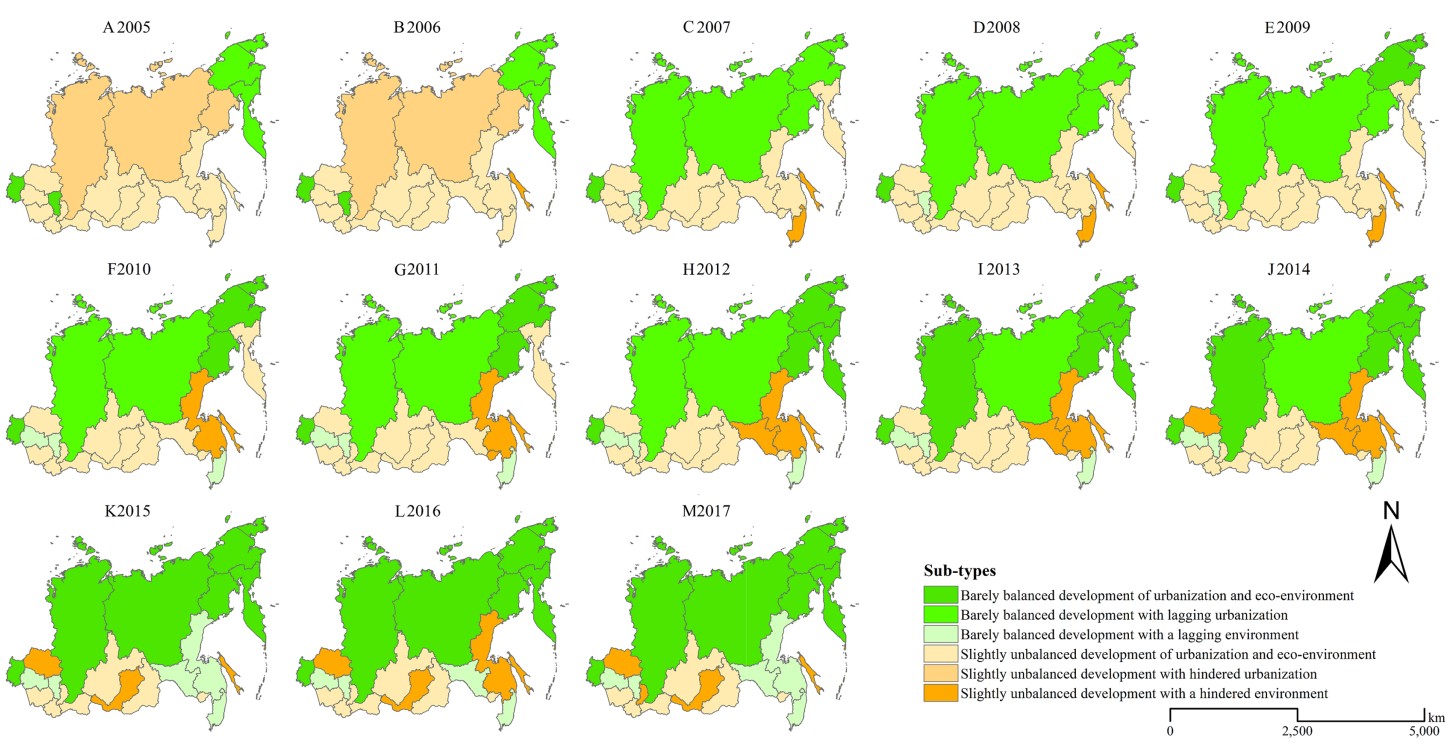

**Figure 7 Spatiotemporal variation characteristics of the coupling coordination degree of urbanization and the ecological environment in the Siberian and Far East Federal Districts.** (A) 2005, (B) 2006, (C) 2007, (D) 2008, (E) 2009, (F) 2010, (G) 2011, (H) 2012, (I) 2013, (J) 2014, (K) 2015, (L) 2016, (M) 2017.

2012, 5 years earlier than the Siberian Federal District. 2009 is a turning point. Since 2009, the coupling coordination degree of urbanization and the ecological environment in the Far East Federal District surpasses the Siberian Federal District, and the difference of the coupling coordination degree of urbanization and the ecological environment between the Siberian Federal District and the Far East Federal District become larger.

From 2005 to 2017, the coupling coordination of urbanization and the ecological environment in the Siberian and Far East Federal Districts improved from slightly unbalanced development stage to barely balanced development stage (shown in Fig. 7). Neither of superior balanced development stage and seriously unbalanced development stage exist in the Siberian and Far East Federal Districts during the study period. The number of the regions, where the coupling coordination development stage of urbanization and the ecological environment achieve barely balanced development during the period 2005–2017, increases from 4 in 2005 to 11 in 2017. In 2005, only 4 regions, including Chukotska Autonomous Oblast and Kamchatskiy Kray in the Far East Federal District, and Republic of Khakassia and Omsk Oblast in the Siberian Federal District, achieved barely balanced development. Since 2007, Republic of Sakha, Magadansk Oblast and Krasnoyarsk Kray have achieved the coupling coordination of urbanization and the ecological environment changing from slightly unbalanced development stage to barely balanced development stage. Since 2010, Primorskiy Kray and Novosibirsk Oblast have achieved the change from slightly unbalanced development stage to barely balanced

development stage. Since 2015, Amursk Oblast and Khabarovskiy Kray have achieved the change from slightly unbalanced development stage to barely balanced development stage. In 2017, more than half regions, including 4 regions in the Siberian Federal District and 7 regions in the Far East Federal District, in the Siberian and Far East Federal Districts achieved the barely balanced development of the coupling coordination of urbanization and the ecological environment.

The spatial pattern of the coupling coordination degree of urbanization and the ecological environment in the Siberian and Far East Federal Districts gradually changes from "dumbbell" pattern to "high-north low-south" pattern. At the early stage of the period 2005–2017, the coupling coordination of urbanization and the ecological environment development stages showed a "dumbbell" spatial pattern. The coupling coordination degree is higher in the east and west regions of the Siberian and Far East Federal Districts, and lower in the central region. At the late stage of the study period, the coupling coordination of urbanization and the ecological environment development stages showed a "high-north low-south" spatial pattern. Only some regions in the south part of the Siberian and Far East Federal Districts have not achieved the balanced development stage.

The coupling coordination degree results of this research are compared with the existing research in China and Mongolia during the recent decade. According to *Yao et al. (2019)* results, the coupling coordination degree of new urbanization and eco-environment stress achieved stable and continuous improvement from 0.389 in 2005 to 0.484 in 2016 in China. Though the stable and continuous increasing trend of the coupling coordination degree in China is similar to the Siberian and Far East Federal Districts in our research, the coupling coordination development of urbanization and the ecological environment in China, staying at the slightly unbalanced development stage and not having achieved the balanced development, is slightly worse than the Siberian and Far East Federal District in Russia Federation. The quantitative assessment of the coupling coordination degree of urbanization and the ecological environment of 22 regions in Mongolia from 2000 to 2016, conducted by *Dong et al. (2019)*, suggests that only two regions, including Ulaanbaatar and Uvurkhangai, are at the slightly unbalanced development stage, and that other regions are all at the seriously unbalanced development in 2005. In 2016, only the capital city Ulaanbaatar achieve the barely balanced development of the coupling coordination of urbanization and the ecological environment, however, more than 80% regions are still at the seriously unbalanced development stage.

## CONCLUSIONS AND DISCUSSION

Based on the comprehensive assessment of the urbanization and the ecological environment, this paper conducts an evaluation of the coupling coordination degree of urbanization and the ecological environment in the Siberian and Far East Federal Districts at regional scale from 2005 to 2017. The temporal variation trend and the spatial pattern of the comprehensive urbanization level, the ecological environment status and the coupling coordination degree are revealed to provide scientific support to achieve the coordinating development of urbanization and the ecological environment in the Siberian

and Far East Federal Districts. Under the background of the Belt and Road Initiative, the results are also meaningful for the green construction of the China–Mongolia–Russia Economic Corridor.

From 2005 to 2017, the comprehensive urbanization level of the Siberian Federal District and the Far East Federal District both keep stable and continuous improvement. The comprehensive urbanization growing rate is higher in the Far East Federal District (6.57%) than the Siberian Federal District (4.12%). Since 2009, the comprehensive urbanization level of the Far East Federal District has surpassed the Siberian Federal District and the gap of the comprehensive urbanization level between the two federal districts become larger. It indicates that the policies of advanced development in the Far East Federal District have important impacts on the socio-economic development of this region. The comprehensive ecological environment level of the Siberian and Far East Federal Districts are similar and keep a relative stable level of 0.20–0.25 during the period 2005–2017.

The coupling coordination of urbanization and the ecological environment in the Siberian and Far East Federal Districts improve from slightly unbalanced development stage to barely balanced development stage from 2005 to 2017. In 2017, more than half regions, including Republic of Khakassia, Omsk Oblast, Novosibirsk Oblast and Krasnoyarsk Kray in the Siberian Federal District and Republic of Sakha, Amursk Oblast, Magadansk Oblast, Chukotska Autonomous Oblast, Khabarovskiy Kray, Primorskiy Kray and Kamchatskiy Kray in the Far East Federal District, achieve the barely balanced development of urbanization and the ecological environment. However, the most desirable development stage, the superior balanced development stage, is never achieved in the Siberian and Far East Federal Districts during the study period. More efforts should be made to achieve the superior balanced development of urbanization and the ecological environment in the study area.

The spatial pattern of the coupling coordination degree of urbanization and the ecological environment in the Siberian and Far East Federal District gradually changes from "dumbbell" to "high-north low-south". At the early stage of the period 2005–2017, the coupling coordination degree of urbanization and the ecological environment is higher in the east and west regions and lower in the central region of the Siberian and Far East Federal Districts. At the late stage, the coupling coordination degree is higher in the north regions and lower in the south regions. In 2017, only some regions in the south part of the Siberian and Far East Federal Districts have not achieved the balanced stage. The south part of the Siberian and Far East Federal Districts should be paid more attention in the future urban development process.

Since the composition differences of the integrated urbanization-ecological environment index system due to the inconsistence of the indicators related to urbanization and the ecological environment in different countries and regions, the comparability of the coupling coordination degree evaluating results among the existing research is limited. Under the background of the Belt and Road Initiative and the construction of the China–Mongolia–Russia Economic Corridor, an integrated evaluation of the coupling coordination degree of urbanization and the ecological environment along

the corridor at the regional scale will be important research topics in the future. Due to the lack of spatial urbanization data, the spatial urbanization indicators are not introduced into the urbanization index subsystem in this paper. Since the ecological environmental data are not available before 2005, this study only evaluates the coupling coordination degree of urbanization and the ecological environment from 2005 to 2017.

### Funding
This work was supported by the Strategic Priority Research Program of the Chinese Academy of Sciences (No. XDA20030203), National Science & Technology Basic Resources Investigation Program (No. 2017FY101303 and No. 2017FY101304). The funders had no role in study design, data collection and analysis, decision to publish, or preparation of the manuscript.

### Grant Disclosures
The following grant information was disclosed by the authors:
Chinese Academy of Sciences: XDA20030203.
National Science & Technology: Nos. 2017FY101303 and 2017FY101304.

### Competing Interests
The authors declare that they have no competing interests.

### Author Contributions
- Ji Zheng conceived and designed the experiments, performed the experiments, analyzed the data, prepared figures and/or tables, authored or reviewed drafts of the paper, and approved the final draft.
- Yingjie Hu performed the experiments, analyzed the data, prepared figures and/or tables, and approved the final draft.
- Tamir Boldanov performed the experiments, analyzed the data, prepared figures and/or tables, authored or reviewed drafts of the paper, and approved the final draft.
- Tcogto Bazarzhapov performed the experiments, analyzed the data, authored or reviewed drafts of the paper, and approved the final draft.
- Dan Meng analyzed the data, authored or reviewed drafts of the paper, and approved the final draft.
- Yu Li conceived and designed the experiments, authored or reviewed drafts of the paper, and approved the final draft.
- Suocheng Dong conceived and designed the experiments, authored or reviewed drafts of the paper, and approved the final draft.

### Data Availability
All raw data of the urbanization-ecological environment are available as a Supplemental File.

## Supplemental Information

Supplemental information for this article can be found online at http://dx.doi.org/10.7717/peerj.9125#supplemental-information.

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
