# Peer review of "Comprehensive assessment of the coupling coordination degree between urbanization and ecological environment in the Siberian and Far East Federal Districts, Russia from 2005 to 2017"

_PeerJ, doi:10.7717/peerj.9125_

## Round 0.1 · original submission · Major Revisions

Based on the comments of the reviewers, I suggest that you make appropriate changes to your article.

Reviewer 1 ·

Basic reporting

The relationship between urbanization and ecological environment is a problem worthy of study. This paper is well organized and provided sufficient analysis,but its novelty needs to be enhanced. And I feel that the literature review is not concise enough in the introduction part.

Experimental design

Methodology section needs to be strengthened. This study employed CCDM to evaluate the coupling coordination degree between urbanization and the ecological environment. I just wonder if the discussion about the methods is enough. It is better to add some contents of the methodologies usually used for this kind of study with the pro and cons. Besides, maybe I missed it, but how did you decide index themselves?

Validity of the findings

The conclusions and discussion part should be enriched and reorganized. It is better to give us more information about the comparison of your research with previous research. You need to compare your results in the context of literature research and discuss their implications.

Reviewer 2 ·

Basic reporting

no comment

Experimental design

no comment

Validity of the findings

no comment

Additional comments

The article is at a very high scientific level. The topic deals with important, practical issues. The theoretical background is well presented. The aim of the article was defined correctly. The research process has been deeply thought through and conducted without objections. Advanced statistics were also used. A reflexive summary was made. The scientific maturity of the authors is confirmed by the presented recommendations.

Reviewer 3 ·

Basic reporting

.

Experimental design

.

Validity of the findings

.

Additional comments

This paper estimated the coupling coordination degree between urbanization and ecological environment in the Siberian and Far East Federal Districts, Russia. The research perspective and content of this study are innovative, and the conclusions are reliable. However, I still have some comments as follows:
1. For the urbanization-ecological environment index system. In the urbanization index system, it includes three first grade indicators (demographic urbanization, economic urbanization, and social urbanization) and 11 basic indicators, while in general, urbanization should also include indicators of spatial urbanization, but why is it not reflected in the article. I suggested to increase indicators in this regard, or give reasons for why not. In addition, there are 11 basic indicators in urbanization index system, but only 6 basic indicators in ecological environment index system, which is unequal and has a big difference in quantity between them, this may lead to inaccuracies when calculate the coupling coordination degree between urbanization and ecological environment.
2. For the results analysis. There are 5 basic indicators in social urbanization, 4 basic indicators in economic urbanization, but only 2 basic indicators in demographic urbanization. The index quantity was unequal among the 3 first grade indicators for urbanization, so that there is no doubt that social urbanization (44.29%) > economic urbanization (33.43%) > demographic urbanization (22.28%), thus this analysis was meaningless.
3. For the discussion. The discussion in the present paper should be improved.

Reviewer 4 ·

Basic reporting

Your introduction gives more research about the relationship between urbanizaion and enviornment, but introduction is not the list most of that research, you should summarize and extact the main thought and progress on the research, and give the gaps or problems of former research.

Experimental design

In your urbanizaiton-ecological environment index system, the basic indicator including total indicators and average indicators. I think some total indicators should change to average indicators, such as Gross regional product, number of high education institutions, number of sports facilities, and all the indicators of ecological environment subsystem.

Validity of the findings

After updating your indicator, you will get different findings. So the results and conclusion maybe need rewrite.

Additional comments

The structure, figures, tables are all good, and the raw data are valuable. Because your key indicators are inappropriate, the paper need rewrite and resubmit.

---

## Round 0.2 · accepted · Accept

After reviewing your revised manuscript of version 1 and the comments from the reviewers, I have decided to accept your article for publication in PeerJ.

Reviewer 1 ·

Basic reporting

no comment

Experimental design

no comment

Validity of the findings

no comment

Additional comments

The revised manuscript has seriously solved my questions with the original manuscript. I think it has met
the publication criteria of Peer J.